# NS$^3$: Neuro-Symbolic Semantic Code Search

**Shushan Arakelyan**[1], **Anna Hakhverdyan**[2], **Miltiadis Allamanis**[3][*],
**Luis Garcia**[4][†], **Christophe Hauser**[4][†], **Xiang Ren**[1][†]
[1]University of Southern California, Department of Computer Science
[2]National Polytechnic University of Armenia
[3]Microsoft Research Cambridge
[4]USC Information Sciences Institute
shushana@usc.edu, annahakhverdyan98@gmail.com
miltos@allamanis.com, {lgarcia, hauser}@isi.edu, xiangren@usc.edu

## Abstract

Semantic code search is the task of retrieving a code snippet given a textual description of its functionality. Recent work has been focused on using similarity metrics between neural embeddings of text and code. However, current language models are known to struggle with longer, compositional text, and multi-step reasoning. To overcome this limitation, we propose supplementing the query sentence with a layout of its semantic structure. The semantic layout is used to break down the final reasoning decision into a series of lower-level decisions. We use a Neural Module Network architecture to implement this idea. We compare our model - NS$^3$ (Neuro-Symbolic Semantic Search) - to a number of baselines, including state-of-the-art semantic code retrieval methods, and evaluate on two datasets - CodeSearchNet and Code Search and Question Answering. We demonstrate that our approach results in more precise code retrieval, and we study the effectiveness of our modular design when handling compositional queries[1].

## 1 Introduction

The increasing scale of software repositories makes retrieving relevant code snippets more challenging. Traditionally, source code retrieval has been limited to keyword [33, 30] or regex [7] search. Both rely on the user providing the exact keywords appearing in or around the sought code. However, neural models enabled new approaches for retrieving code from a textual description of its functionality, a task known as *semantic code search (SCS)*. A model like Transformer [36] can map a database of code snippets and natural language queries to a shared high-dimensional space. Relevant code snippets are then retrieved by searching over this embedding space using a predefined similarity metric, or a learned distance function [26, 13, 12]. Some of the recent works capitalize on the rich structure of the code, and employ graph neural networks for the task [17, 28].

Despite impressive results on SCS, current neural approaches are far from satisfactory in dealing with a wide range of natural-language queries, especially on ones with compositional language structure. Encoding text into a dense vector for retrieval purposes can be problematic because we risk loosing faithfulness of the representation, and missing important details of the query. Not only does this a) affect the performance, but it can b) drastically reduce a model's value for the users, because compositional queries such as "*Check that directory does not exist before creating it*" require performing multi-step reasoning on code.

---

[*]Currently at Google Research
[†]Equal supervision
[1]Code and data are available at https://github.com/ShushanArakelyan/modular_code_search

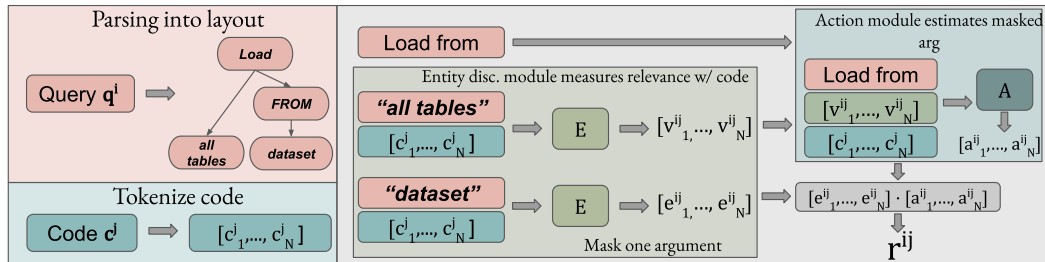

Figure 2: **Overview of the NS3 approach**. We illustrate the pipeline of processing for an example query "*Load all tables from dataset*". Parsed query is used for deciding the positions of entity discovery and action modules in the neural module network layout. Each entity discovery module receives a noun/noun phrase as input, and outputs relatedness scores for code tokens, which are passed as input to an action module. Action module gets scores for all its children in the parse-tree, except one, which is masked, and the goal is predicting, cloze-style, what are the relatedness scores for the missing argument.

We suggest overcoming these challenges by introducing a modular workflow based on the semantic structure of the query. Our approach is based on the intuition of how an engineer would approach a SCS task. For example, in performing search for code that navigates folders in Python they would first only pay attention to code that has cues about operating with paths, directories or folders. Afterwards, they would seek indications of iterating through some of the found objects or other entities in the code related to them. In other words, they would perform multiple steps of different nature - i.e. finding indications of specific types of data entities, or specific operations. Figure 1 illustrates which parts of the code would be

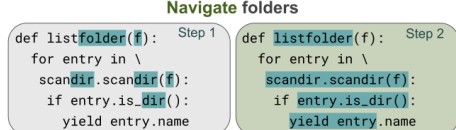

Figure 1: **Motivating Example**. To match query "*Navigate folders*" on a code snippet, we find all references (token spans) to entity "*folders*" in code (e.g., paths and directories) using various linguistic cues (Step 1). Then we look for cues in code that indicate the identified instances of "*folders*" are being iterated through – i.e., "*navigate*" (Step 2).

important to indicate that they have found the desired code snippet at each step. We attempt to imitate this process in this work. To formalize the decomposition of the query into such steps, we take inspiration from the idea that code is comprised of data, or entities, and transformations, or actions, over data. Thus, a SCS query is also likely to describe the code in terms of data entities and actions.

We break down the task of matching the query into smaller tasks of matching individual data entities and actions. In particular, we aim to identify parts of the code that indicate the presence of the corresponding data or action. We tackle each part with a distinct type of network – a neural module. Using the semantic parse of the query, we construct the layout of how modules' outputs should be linked according to the relationships between data entities and actions, where each data entity represents a noun, or a noun phrase, and each action represents a verb, or a verbal phrase. Correspondingly, this layout specifies how the modules should be combined into a single neural module network (NMN) [4]. Evaluating the NMN on the candidate code approximates detecting the corresponding entities and actions in the code by testing whether the neural network can deduce one missing entity from the code and the rest of the query.

This approach has the following advantages. First, semantic parse captures the compositionality of a query. Second, it mitigates the challenges of faithful encoding of text by focusind only on a small portion of the query at a time. Finally, applying the neural modules in a succession can potentially mimic staged reasoning necessary for SCS.

We evaluate our proposed NS³ model on two SCS datasets - CodeSearchNet (CSN) [24] and CoSQA/WebQueryTest [23]. Additionally, we experiment with a limited training set size of CSN of 10K and 5K examples. We find that NS³ provides large improvements upon baselines in all cases. Our experiments demonstrate that the resulting model is more sensitive to small, but semantically significant changes in the query, and is more likely to correctly recognize that a modified query no longer matches its code pair.

Our main contributions are: (i) We propose looking at SCS as a compositional task that requires multi-step reasoning. (ii) We present an implementation of the aforementioned paradigm based on

NMNs. (iii) We demonstrate that our proposed model provides a large improvement on a number of well-established baseline models. (iv) We perform additional studies to evaluate the capacity of our model to handle compositional queries.

## 2 Background

### 2.1 Semantic Code Search

Semantic code search (SCS) is the process of retrieving a relevant code snippet based on a textual description of its functionality, also referred to as *query*. Let $\mathcal{C}$ be a database of code snippets $\mathbf{c}^i$. For each $\mathbf{c}^i \in \mathcal{C}$, there is a textual description of its functionality $\mathbf{q}^i$. In the example in Figure 2, the query $\mathbf{q}^i$ is "*Load all tables from dataset*". Let $r$ be an indicator function such that $r(\mathbf{q}^i, \mathbf{c}^j) = 1$ if $i = j$; and 0 otherwise. Given some query $\mathbf{q}$ the goal of SCS is to find $\mathbf{c}^*$ such that $r(\mathbf{q}, \mathbf{c}^*) = 1$. We assume that for each $\mathbf{q}^*$ there is exactly one such $\mathbf{c}^*$.[2] Here we look to construct a model which takes as input a pair of query and a candidate code snippet: $(\mathbf{q}^i, \mathbf{c}^j)$ and assign the pair a probability $\hat{r}^{ij}$ for being a correct match. Following the common practice in information retrieval, we evaluate the performance of the model based on how high the correct answer $\mathbf{c}^*$ is ranked among a number of incorrect, or distractor instances $\{\mathbf{c}\}$. This set of distractor instances can be the entire codebase $\mathcal{C}$, or a subset of the codebase obtained through heuristic filtering, or another ranking method.

### 2.2 Neural Models for Semantic Code Search

Past works handling programs and code have focused on enriching their models with incorporating more semantic and syntactic information from code [1, 10, 34, 47]. Some prior works have cast the SCS as a sequence classification task, where the code is represented as a textual sequence and input pair $(\mathbf{q}^i, \mathbf{c}^j)$ is concatenated with a special separator symbol into a single sequence, and the output is the score $\hat{r}^{ij}$: $\hat{r}^{ij} = f(\mathbf{q}^i, \mathbf{c}^j)$. The function $f$ performing the classification can be any sequence classification model, e.g. BERT [11].

Alternatively, one can define separate networks for independently representing the query ($f$), the code ($g$) and measuring the similarity between them: $\hat{r}^{ij} = sim(f(\mathbf{q}^i), g(\mathbf{c}^j))$. This allows one to design the code encoding network $g$ with additional program-specific information, such as abstract syntax trees [3, 44] or control flow graphs [15, 45]. Separating two modalities of natural language and code also allows further enrichment of code representation by adding contrastive learning objectives [25, 6]. In these approaches, the original code snippet $\mathbf{c}$ is automatically modified with semantic-preserving transformations, such as variable renaming, to introduce versions of the code snippet - $\mathbf{c}'$ with the exact same functionality. Code encoder $g$ is then trained with an appropriate contrastive loss, such as Noise Contrastive Estimation (NCE) [19], or InfoNCE [35].

**Limitations** However, there is also merit in reviewing how we represent and use the textual *query* to help guide the SCS process. Firstly, existing work derives a single embedding for the entire query. This means that specific details or nested subqueries of the query may be omitted or not represented faithfully - getting lost in the embedding. Secondly, prior approaches make the decision after a single pass over the code snippet. This ignores cases where reasoning about a query requires multiple steps and thus - multiple look-ups over the code, as is for example in cases with nested subqueries. Our proposed approach - $NS^3$ - attempts to address these issues by breaking down the query into smaller phrases based on its semantic parse and locating each of them in the code snippet. This should allow us to match compositional and longer queries to code more precisely.

## 3 Neural Modular Code Search

We propose to supplement the query with a loose structure resembling its semantic parse, as illustrated in Figure 2. We follow the parse structure to break down the query into smaller, semantically coherent parts, so that each corresponds to an individual execution step. The steps are taken in succession by a neural module network composed from a layout that is determined from the semantic parse of the

---

[2]This is not the case in CoSQA dataset. For the sake of consistency, we perform the evaluation repeatedly, leaving only one correct code snippet among the candidates at a time, while removing the others.

query (Sec. 3.1). The neural module network is composed by stacking "modules", or jointly trained networks, of distinct types, each carrying out a different functionality.

**Method Overview** In this work, we define two types of neural modules - *entity discovery* module (denoted by $E$; Sec. 3.2) and *action* module (denoted by $A$; Sec 3.3). The entity discovery module estimates semantic relatedness of each code token $c_i^j$ in the code snippet $\mathbf{c}^j = [c_1^j, \ldots, c_N^j]$ to an entity mentioned in the query – e.g. "*all tables*" or "*dataset*" as in Figure 2. The action module estimates the likelihood of each code token to be related to an (unseen) entity affected by the action in the query e.g. "*dataset*" and "*load from*" correspondingly, conditioned on the rest of the input (seen), e.g. "*all tables*". The similarity of the predictions of the entity discovery and action modules measures how well the code matches that part of the query. The modules are nested - the action modules are taking as input part of the output of another module - and the order of nesting is decided by the semantic parse layout. In the rest of the paper we refer to the inputs of a module as its *arguments*.

Every input instance fed to the model is a 3-tuple $(\mathbf{q}^i, s_{q^i}, \mathbf{c}^j)$ consisting of a natural language query $\mathbf{q}^i$, the query's semantic parse $s_{q^i}$, a candidate code (sequence) $\mathbf{c}^j$. The goal is producing a binary label $\hat{r}^{ij} = 1$ if the code is a match for the query, and 0 otherwise. The layout of the neural module network, denoted by $L(s_{q^i})$, is created from the semantic structure of the query $s_{q^i}$. During inference, given $(\mathbf{q}^i, s_{q^i}, \mathbf{c}^j)$ as input the model instantiates a network based on the layout, passes $\mathbf{q}^i$, $\mathbf{c}^j$ and $s_{q^i}$ as inputs, and obtains the model prediction $\hat{r}^{ij}$. This pipeline is illustrated in Figure 2, and details about creating the layout of the neural module network are presented in Section 3.1.

During training, we first perform noisy supervision pretraining for both modules. Next, we perform end-to-end training, where in addition to the query, its parse, and a code snippet, the model is also provided a gold output label $r(\mathbf{q}^i, \mathbf{c}^j) = 1$ if the code is a match for the query, and $r(\mathbf{q}^i, \mathbf{c}^j) = 0$ otherwise. These labels provide signal for joint fine-tuning of both modules (Section 3.5).

## 3.1 Module Network Layout

Here we present our definition of the structural representation $s_{q^i}$ for a query $\mathbf{q}^i$, and introduce how this structural representation is used for dynamically constructing the neural module network, i.e. building its layout $L(s_{q^i})$.

**Query Parsing** To infer the representation $s_{q^i}$, we pair the query (*e.g.*, "*Load all tables from dataset*", as in Figure 2), with a simple semantic parse that looks similar to: `DO WHAT [ (to/from/in/...) WHAT, WHEN, WHERE, HOW, etc]`. Following this semantic parse, we break down the query into shorter semantic phrases using the roles of different parts of speech. Nouns and noun phrases correspond to data entities in code, and verbs describe actions or transformations performed on the data entities. Thus, data and transformations are separated and handled by separate neural modules – an *entity discovery module E* and an *action module A*. We use a Combinatory Categorial Grammar-based (CCG) semantic parser [43, 5] to infer the semantic parse $s_{q^i}$ for the natural language query $\mathbf{q}^i$. Parsing is described in further detail in Section 4.1 and Appendix A.2.

**Specifying Network Layout** In the layout $L(s_{q^i})$, every noun phrase (e.g., "*dataset*" in Figure 2) will be passed through the entity discovery module $E$. Module $E$ then produces a probability score $e_k$ for every token $c_k^j$ in the code snippet $\mathbf{c}^j$ to indicate its semantic relatedness to the noun phrase: $E(\text{"}dataset\text{"}, \mathbf{c}^j) = [e_1, e_2, \ldots, e_N]$. Each verb in $s_{q^i}$ (e.g., "*load*" in Figure 2) will be passed through an action module: $A(\text{"}load\text{"}, \mathbf{p}^i, \mathbf{c}^j) = [a_1, a_2, \ldots, a_N]$. Here, $\mathbf{p}^i$ is the span of arguments to the verb (action) in query $\mathbf{q}^i$, consisting of children of the verb in the parse $s_{q^i}$ (e.g. subject and object arguments to the predicate "*load*"); $a_1, \ldots, a_N$ are estimates of the token scores $e_1, \ldots, e_N$ for an entity from $\mathbf{p}^i$. The top-level of the semantic parse is always an action module. Figure 2 also illustrates preposition FROM used with "*dataset*", handling which is described in Section 3.3.

## 3.2 Entity Discovery Module

The entity discovery module receives a string that references a data entity. Its goal is to identify tokens in the code that have high relevance to that string. The architecture of the module is shown in Figure 3. Given an entity string, "*dataset*" in the example, and a sequence of code tokens $[c_1^j, \ldots, c_N^j]$, entity module first obtains contextual code token representation using RoBERTa model that is initialized

from CodeBERT-base checkpoint. The resulting embedding is passed through a two-layer MLP to obtain a score for every individual code token $c_k^j : 0 \leq e_k \leq 1$. Thus, the total output of the module is a vector of scores: $[e_1, e_2, \ldots, e_N]$. To prime the entity discovery module for measuring relevancy between code tokens and input, we fine-tune it with noisy supervision, as detailed below.

**Noisy Supervision** We create noisy supervision for the entity discovery module by using keyword matching and a Python static code analyzer. For the keyword matching, if a code token is an exact match for one or more tokens in the input string, its supervision label is set to 1, otherwise it is 0. Same is true if the code token is a substring or a superstring of one or more input string tokens. For some common nouns we include their synonyms (e.g. "*map*" for "*dict*"), the full list of those and further details are presented in Appendix B.

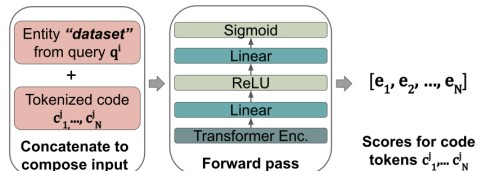

Figure 3: Entity module architecture.

We used the static code analyzer to extract information about statically known data types. We cross-matched this information with the query to discover whether the query references any datatypes found in the code snippet. If that is the case, the corresponding code tokens are assigned supervision label 1, and all the other tokens are assigned to 0. In the pretraining we learned on equal numbers of $(query, code)$ pairs from the dataset, as well as randomly mismatched pairs of queries and code snippets to avoid creating bias in the entity discovery module.

### 3.3 Action Module

First, we discuss the case where the action module has only entity module inputs. Figure 4 provides a high-level illustration of the action module. In the example, for the query "*Load all tables from dataset*", the action module receives only part of the full query – "*Load all tables from ???*". Action module then outputs token scores for the masked argument – "*dataset*". If the code snippet corresponds to the query, then the action module should be able to deduce this missing part from the code and the rest of the query. For consistency, we always mask the last data entity argument. We pre-train the action module using the output scores of the entity discovery module as supervision.

Each data entity argument can be associated with 0 or 1 prepositions, but each action may have multiple entities with prepositions. For that reason, for each data entity argument we create one joint embedding of the action verb and the preposition. Joint embeddings are obtained with a 2-layer MLP model, as illustrated in the left-most part of Figure 4.

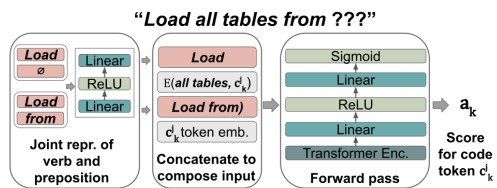

If a data entity does not have a preposition associated with it, the vector corresponding to the preposition is filled with zeros. The joint verb-preposition embedding is stacked with the code token embedding $c_k^j$ and entity discovery module output for that token, this is referenced in the middle part of Figure 4. This vector is passed through a transformer encoder model, followed by a 2-layer MLP and a

Figure 4: Action module architecture.

sigmoid layer to output token score $a_k$, illustrated in the right-most part of the Figure 4. Thus, the dimensionality of the input depends on the number of entities. We use a distinct copy of the module with the corresponding dimensionality for different numbers of inputs, from 1 to 3.

### 3.4 Model Prediction

The final score $\hat{r}^{ij} = f(\mathbf{q}^i, \mathbf{c}^j)$ is computed based on the similarity of action and entity discovery module output scores. Formally, for an action module with verb $x$ and parameters $\mathbf{p}^x = [p_1^x, \ldots, p_k^x]$, the final model prediction is the dot product of respective outputs of action and entity discovery modules: $\hat{r}^{ij} = A(x, p_1^x, \ldots, p_{k-1}^x) \cdot E(p_k^x)$. Since the action module estimates token scores for the entity affected by the verb, if its prediction is far from the truth - then either the action is not found in the code, or it is not fully corresponding to the query, for example, in the code snippet tables are loaded from web, instead of a dataset. We normalize this score to make it a probability. If this is the

only action in the query, this probability score will be the output of the entire model for $(\mathbf{q}^i, \mathbf{c}^j)$ pair: $\hat{r}^{ij}$, otherwise $\hat{r}^{ij}$ will be the product of probability scores of all nested actions in the layout.

**Compositional query with nested actions**    Consider a compositional query "*Load all tables from dataset using* Lib *library*". Here action with verb "*Load from*" has an additional argument "*using*" – also an action – with an entity argument "Lib *library*". In case of nested actions, we flatten the layout by taking the conjunction of individual action similarity scores. Formally, for two verbs $x$ and $y$ and their corresponding arguments $\mathbf{p}^x = [p_1^x, \ldots, p_k^x]$ and $\mathbf{p}^y = [p_1^y, \ldots, p_l^y]$ in a layout that looks like: $A(x, \mathbf{p}^x, A(y, \mathbf{p}^y))$, the output of the model is the conjunction of similarity scores computed for individual action modules: $sim(A(x, p_1^x, \ldots, p_{k-1}^x), E(p_k^x)) \cdot sim(A(y, p_1^y, \ldots, p_{l-1}^y), E(p_l^y))$. This process is repeated until all remaining $\mathbf{p}^x$ and $\mathbf{p}^y$ are data entities. This design ensures that code snippet is ranked highly if both actions are ranked highly, we leave explorations of alternative handling approaches for nested actions to future work.

## 3.5   Module Pretraining and Joint Fine-tuning

We train our model through supervised pre-training, as is discussed in Sections 3.2 and 3.3, followed by end-to-end training. End-to-end training objective is binary classification - given a pair of query $\mathbf{q}^i$ and code $\mathbf{c}^j$, the model predicts probability $\hat{r}^{ij}$ that they are related. In the end-to-end training, we use positive examples taken directly from the dataset - $(\mathbf{q}^i, \mathbf{c}^i)$, as well as negative examples composed through the combination of randomly mismatched queries and code snippets. The goal of end-to-end training is fine-tuning parameters of entity discovery and action modules, including the weights of the RoBERTA models used for code token representation.

Batching is hard to achieve for our model, so for the interest of time efficiency we do not perform inference on all distractor code snippets in the code dataset. Instead, for a given query we re-rank top-K highest ranked code snippets as outputted by some baseline model, in our evaluations we used CodeBERT. Essentially, we use our model in a re-ranking setup, this is common in information retrieval and is known as L2 ranking. We interpret the probabilities outputted by the model as ranking scores. More details about this procedure are provided in Section 4.1.

# 4   Experiments

## 4.1   Experiment Setting

**Dataset**    We conduct experiments on two datasets: Python portion of the **CodeSearchNet** (CSN) [24], and **CoSQA** [23]. We parse all queries with the CCG parser, as discussed later in this section, excluding unparsable examples from further experiments. This leaves us with approximately 40% of the CSN dataset and 70% of the CoSQA dataset, the exact data statistics are available in Appendix A in Table 3. We believe, that the difference in success rate of the parser between the two datasets can be attributed to the fact that CSN dataset, unlike CoSQA, does not contain real code search queries, but rather consists of docstrings, which are used as approximate queries. More details and examples can be found in Appendix A.3. For our baselines, we use the parsed portion of the dataset for fine-tuning to make the comparison fair. In addition, we also experiment with fine-tuning all models on an even smaller subset of CodeSearchNet dataset, using only 5K and 10K examples for fine-tuning. The goal is testing whether modular design makes NS3 more sample-efficient.

All experiment and ablation results discussed in Sections 4.2, 4.3 and 4.4 are obtained on the test set of CSN for models trained on CSN training data, or WebQueryTest [31] – a small natural language web query dataset of document-code pairs – for models trained on CoSQA dataset.

**Evaluation and Metrics**    We follow CodeSearchNet's original approach for evaluation for a test instance $(q, c)$, comparing the output against outputs over a fixed set of 999 distractor code snippets. We use two evaluation metrics: Mean Reciprocal Rank (MRR) and Precision@K (P@K) for K=1, 3, and 5, see Appendix A.1 for definitions and further details.

Following a common approach in information retrieval, we perform two-step evaluation. In the first step, we obtain CodeBERT's output against 999 distractors. In the second step, we use $NS^3$ to re-rank the top 10 predictions of CodeBERT. This way the evaluation is much faster, since unlike our

| Method | CSN | | | | CSN-10K | | | | CSN-5K | | | |
|---|---|---|---|---|---|---|---|---|---|---|---|---|
| | MRR | P@1 | P@3 | P@5 | MRR | P@1 | P@3 | P@5 | MRR | P@1 | P@3 | P@5 |
| BM25 | 0.209 | 0.144 | 0.230 | 0.273 | 0.209 | 0.144 | 0.230 | 0.273 | 0.209 | 0.144 | 0.230 | 0.273 |
| RoBERTa (code) | 0.842 | 0.768 | 0.905 | 0.933 | 0.461 | 0.296 | 0.545 | 0.664 | 0.290 | 0.146 | 0.324 | 0.438 |
| CuBERT | 0.225 | 0.168 | 0.253 | 0.294 | 0.144 | 0.081 | 0.166 | 0.214 | 0.081 | 0.030 | 0.078 | 0.118 |
| CodeBERT | 0.873 | 0.803 | 0.939 | 0.958 | 0.69 | 0.55 | 0.799 | 0.873 | 0.680 | 0.535 | 0.794 | 0.870 |
| GraphCodeBERT | 0.812 | 0.725 | 0.880 | 0.919 | 0.786 | 0.684 | 0.859 | 0.901 | 0.773 | 0.677 | 0.852 | 0.892 |
| GraphCodeBERT* | 0.883 | 0.820 | 0.941 | 0.962 | 0.780 | 0.683 | 0.858 | 0.904 | 0.765 | 0.662 | 0.846 | 0.894 |
| NS$^3$ | **0.924** | **0.884** | **0.959** | **0.969** | **0.826** | **0.753** | **0.886** | **0.908** | **0.823** | **0.751** | **0.881** | **0.913** |
| Upper-bound | 0.979 | | | | 0.939 | | | | 0.936 | | | |

Table 1: Mean Reciprocal Rank (MRR) and Precision@1/@3/@5 (higher is better) for methods trained on different subsets from CodeSearchNet dataset.

modular approach, CodeBERT can be fed examples in batches. And as we will see from the results, we see improvement in final performance in all scenarios.

**Compared Methods**   We compare NS$^3$ with various state-of-the-art methods, including some traditional approaches for document retrieval and pretrained large NLP language models. (1) **BM25** is a ranking method to estimate the relevance of documents to a given query. (2) **RoBERTa (code)** is a variant of RoBERTa [29] pretrained on the CodeSearchNet corpus. (3) **CuBERT** [26] is a BERT Large model pretrained on 7.4M Python files from GitHub. (4) **CodeBERT** [13] is an encoder-only Transformer model trained on unlabeled source code via masked language modeling (MLM) and replaced token detection objectives. (5) **GraphCodeBERT** [17] is a pretrained Transformer model using MLM, data flow edge prediction, and variable alignment between code and the data flow. (6) **GraphCodeBERT\*** is a re-ranking baseline. We used the same setup as for NS3, but used GraphCodeBERT to re-rank the top-10 predictions of the CodeBERT model.

**Query Parser**   We started by building a vocabulary of predicates for common action verbs and entity nouns, such as "*convert*", "*find*", "*dict*", "*map*", etc. For those we constructed the lexicon (rules) of the parser. We have also included "catch-all" rules, for parsing sentences with less-common words. To increase the ratio of the parsed data, we preprocessed the queries by removing preceding question words, punctuation marks, etc. Full implementation of our parser including the entire lexicon and vocabulary can be found at `https://anonymous.4open.science/r/ccg_parser-4BC6`. More details are available in Appendix A.2.

| Method | CoSQA | | | |
|---|---|---|---|---|
| | MRR | P@1 | P@3 | P@5 |
| BM25 | 0.103 | 0.05 | 0.119 | 0.142 |
| RoBERTa (code) | 0.279 | 0.159 | 0.343 | 0.434 |
| CuBERT | 0.127 | 0.067 | 0.136 | 0.187 |
| CodeBERT | 0.345 | 0.175 | 0.42 | 0.54 |
| GraphCodeBERT | 0.435 | 0.257 | 0.538 | 0.628 |
| GraphCodeBERT* | 0.462 | 0.314 | 0.547 | 0.632 |
| NS$^3$ | **0.551** | **0.445** | **0.619** | **0.668** |
| Upper-bound | 0.736 | 0.724 | 0.724 | 0.724 |

Table 2: Mean Reciprocal Rank(MRR) and Precision@1/@3/@5 (higher is better) for different methods trained on CoSQA dataset.

**Pretrained Models**   Action and entity discovery modules each embed code tokens with a RoBERTa model, that has been initialized from a checkpoint of pretrained CodeBERT model [3]. We fine-tune these models during the pretraining phases, as well as during final end-to-end training phase.

**Hyperparameters**   The MLPs in entity discovery and action modules have 2 layers with input dimension of 768. We use dropout in these networks with rate 0.1. The learning rate for pretraining and end-to-end training phases was chosen from the range of 1e-6 to 6e-5. We use early stopping with evaluation on unseen validation set for model selection during action module pretraining and end-to-end training. For entity discovery model selection we performed manual inspection of produced scores on unseen examples. For fine-tuning the CuBERT, CodeBERT and GraphCodeBERT baselines we use the hyperparameters reported in their original papers. For RoBERTa (code), we perform the search for learning rate during fine-tuning stage in the same interval as for our model. For model selection on baselines we also use early stopping.

---

[3] `https://huggingface.co/microsoft/codebert-base`

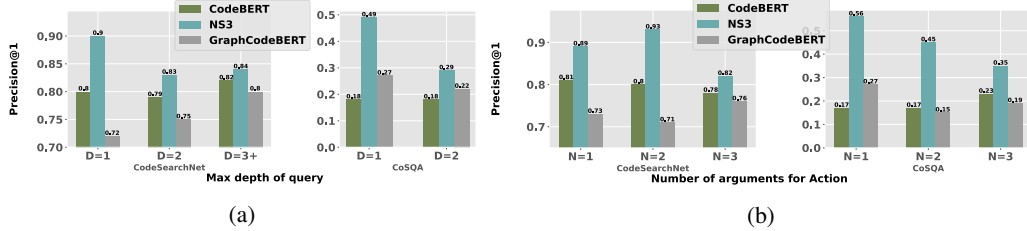

(a)                                           (b)

Figure 5: We report Precision@1 scores. (a) Performance of our proposed method and baselines broken down by average number of arguments per action in a single query. (b) Performance of our proposed method and baselines broken down by number of arguments in queries with a single action.

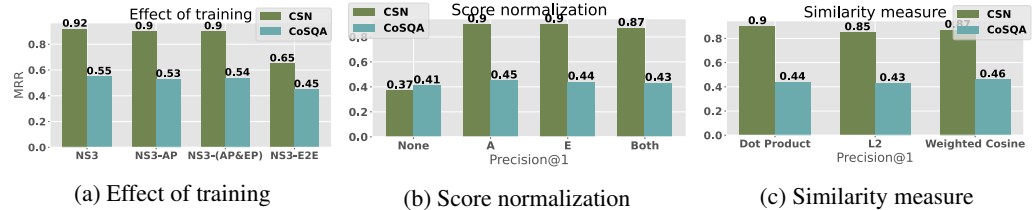

(a) Effect of training          (b) Score normalization          (c) Similarity measure

Figure 6: Performance of $NS^3$ on the test portion of CSN dataset with different ablation variants. (a) Skipping one, or both pretraining procedures, and only training end-to-end. (b) Using no normalization on output scores (**None**), action-only or entity discovery-only, and both. (c) Performance with different options for computing action and entity discovery output similarities.

## 4.2   Results

**Performance Comparison**   Tables 1 and 2 present the performance evaluated on testing portion of CodeSearchNet dataset, and WebQueryTest dataset correspondingly. As it can be seen, our proposed model outperforms the baselines.

Our evaluation strategy improves performance only if the correct code snippet was ranked among the top-10 results returned by the CodeBERT model, so rows labelled "Upper-bound" report best possible performance with this evaluation strategy.

**Query Complexity vs. Performance**   Here we present the breakdown of the performance for our method vs baselines, using two proxies for the complexity and compositionality of the query. The first one is the maximum depth of the query. We define the maximum depth as the maximum number of nested action modules in the query. The results for this experiment are presented in Figure 5a. As we can see, $NS^3$ improves over the baseline in all scenarios. It is interesting to note, that while CodeBERT achieves the best performance on queries with depth 3+, our model's performance peaks at depth = 1. We hypothesize that this can be related to the automated parsing procedure, as parsing errors are more likely to be propagated in deeper queries. Further studies with carefully curated manual parses are necessary to better understand this phenomenon.

Another proxy for the query complexity we consider, is the number of data arguments to a single action module. While the previous scenario is breaking down the performance by the depth of the query, here we consider its "width". We measure the average number of entity arguments per action module in the query. In the parsed portion of our dataset we have queries that range from 1 to 3 textual arguments per action verb. The results for this evaluation are presented in Figure 5. As it can be seen, there is no significant difference in performances between the two groups of queries in either CodeBERT or our proposed method - $NS^3$.

## 4.3   Ablation Studies

**Effect of Pretraining**   In an attempt to better understand the individual effect of the two modules as well as the roles of their pretraining and training procedures, we performed two additional ablation studies. In the first one, we compare the final performance of the original model with two versions where we skipped part of the pretraining. The model noted as $(NS^3 - AP)$ was trained with pretrained entity discovery module, but no pretraining was done for action module, instead we proceeded to the end-to-end training directly. For the model called $NS^3 - (AP\&EP)$, we skipped both pretrainings

of the entity and action modules, and just performed end-to-end training. Figure 6a demonstrates that combined pretraining is important for the final performance. Additionally, we wanted to measure how effective the setup was without end-to-end training. The results are reported in Figure 6a under the name $NS^3 - E2E$. There is a huge performance dip in this scenario, and while the performance is better than random, it is obvious that end-to-end training is crucial for $NS^3$.

**Score Normalization**    We wanted to determine the importance of output normalization for the modules to a proper probability distribution. In Figure 6b we demonstrate the performance achieved using no normalization at all, normalizing either action or entity discovery module, or normalizing both. In all cases we used L1 normalization, since our output scores are non-negative. The version that is not normalized at all performs the worst on both datasets. The performances of the other three versions are close on both datasets.

**Similarity Metric**    Additionally, we experimented with replacing the dot product similarity with a different similarity metric. In particular, in Figure 6c we compare the performance achieved using dot product similarity, L2 distance, and weighted cosine similarity. The difference in performance among different versions is marginal.

### 4.4    Analysis and Case Study

Appendix C contains additional studies on model generalization, such as handling completely unseen actions and entities, as well as the impact of the frequency of observing an action or entity during training has on model performance.

**Case Study**    Finally, we demonstrate some examples of the scores produced by our modules at different stages of training. Figure 8 shows module score outputs for two different queries and with their corresponding code snippets. The first column shows the output of the entity discovery module after pretraining, while the second and third columns demonstrate the outputs of entity discovery and action modules after the end-to-end training. We can see that in the first column the model identifies syntactic matches, such as "folder" and a list comprehension, which "elements" could be related too. After fine-tuning we can see there is a wider range of both syntactic and some semantic matches present, e.g. "dirlist" and "filelist" are correctly identified as related to "folders".

**Perturbed Query Evaluation**    In this section we study how sensitive the models are to small changes in the query $\mathbf{q}^i$, so that it no longer correctly describes its corresponding code snippet $\mathbf{c}^i$. Our expectation is that evaluating a sensitive model on $\mathbf{c}^i$ will rate the original query higher than the perturbed one. Whereas a model that tends to over-generalize and ignore details of the query will likely rate the perturbed query similar to the original. We start from 100 different pairs $(\mathbf{q}^i, \mathbf{c}^i)$, that both our model and CodeBERT predict correctly.

We limited our study to queries with a single verb and a single data entity argument to that verb. For each pair we generated perturbations of two kinds, with 20 perturbed versions for every query. For the first type of perturbations, we replaced query's data argument with a data argument sampled randomly from another query. For the second type, we replaced the verb argument with another randomly sampled verb. To account for calibration of the models, we measure the change in performance through ratio of the perturbed query score over original query score (lower is better). The results are shown in Figure 7. labelled "$V(arg_1) \rightarrow V(arg_2)$" and "$V_1(arg) \rightarrow V_2(arg)$".

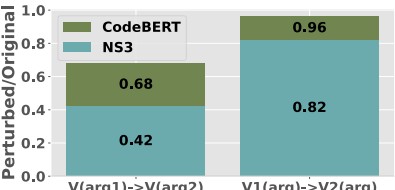

Figure 7: Ratio of the perturbed query score to the original query score (lower is better) on CSN dataset.

**Discussion**    One of the main requirements for the application of our proposed method is being able to construct a semantic parse of the retrieval query. In general, it is reasonable to expect the users of the SCS to be able to come up with a formal representation of the query, e.g. by representing it in a form similar to SQL or CodeQL. However, due to the lack of such data for training and testing purposes, we implemented our own parser, which understandably does not have perfect performance since we are dealing with open-ended sentences.

```
Navigate        def listfolder(p):        def listfolder(p):        def listfolder(p):
folders           for entry in \           for entry in \           for entry in \
              scandir.scandir(p):      scandir.scandir(p):      scandir.scandir(p):
            if entry.is_dir():       if entry.is_dir():       if entry.is_dir():
                yield entry.name         yield entry.name         yield entry.name

Remove          def dedup_list(l):       def dedup_list(l):       def dedup_list(l):
redundant         dedup = set()            dedup = set()            dedup = set()
elements          return [x for x in l \   return [x for x in l \   return [x for x in l \
of list             if not (x in dedup \     if not (x in dedup \     if not (x in dedup \
                  or dedup.add(x))]        or dedup.add(x))]        or dedup.add(x))]

            Entity after pretraining   Entity after finetuning   Action after finetuning
```

Figure 8: Token scores outputted by the modules at different stages of training. Darker highlighting means higher score. The leftmost and middle columns show output scores of the entity discovery module after pretraining, and the end-to-end training correspondingly. The rightmost column shows the scores of the action module after the end-to-end training.

## 5 Related work

Different deep learning models have proved quite efficient when applying to programming languages and code. Prior works have studied and reviewed the uses of deep learning for code analysis in general and code search in particular [39, 31].

A number of approaches to deep code search is based on creating a relevance-predicting model between text and code. [16] propose using RNNs for embedding both code and text to the same latent space. On the other hand, [27] capitalizes the inherent graph-like structure of programs to formulate code search as graph matching. A few works propose enriching the models handling code embedding by adding additional code analysis information, such as semantic and dependency parses [12, 2], variable renaming and statement permutation [14], as well as structures such as abstract syntax tree of the program [20, 37]. A few other approaches have dual formulations of code retrieval and code summarization [9, 40, 41, 6] In a different line of work, Heyman & Cutsem [21] propose considering the code search scenario where short annotative descriptions of code snippets are provided. Appendix E discusses more related work.

## 6 Conclusion

We presented NS$^3$ a symbolic method for semantic code search based on neural module networks. Our method represents the query and code in terms of actions and data entities, and uses the semantic structure of the query to construct a neural module network. In contrast to existing code search methods, NS$^3$ more precisely captures the nature of queries. In an extensive evaluation, we show that this method works better than strong but unstructured baselines. We further study model's generalization capacities, robustness, and sensibility of outputs in a series of additional experiments.

### Acknowledgments and Disclosure of Funding

This research is supported in part by the DARPA ReMath program under Contract No. HR00112190020, the DARPA MCS program under Contract No. N660011924033, Office of the Director of National Intelligence (ODNI), Intelligence Advanced Research Projects Activity (IARPA), via Contract No. 2019-19051600007, the Defense Advanced Research Projects Agency with award W911NF-19-20271, NSF IIS 2048211, and gift awards from Google, Amazon, JP Morgan and Sony. The views and conclusions contained herein are those of the authors and should not be interpreted as necessarily representing the official policies or endorsements, either expressed or implied, of DARPA, ODNI, IARPA, or the U.S. Government. The U.S. Government is authorized to reproduce and distribute reprints for Governmental purposes notwithstanding any copyright annotation thereon. We thank all the collaborators in USC INK research lab for their constructive feedback on the work.

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
