# Appendices for Submission # 2981

Below we include additional implementation details, experimental results, as well as findings and analyses. The code implementing the model is included in the supplementary materials folder. Section A details our setup and evaluation, providing additional information on evaluation metrics, dataset statistics and CCG parser. Section B discusses implementation details of the entity discovery module. Section C contains additional experiments, where the performance is broken down by the frequency of appearance of tokens in the training data, including break-down over unseen tokens. Section D has some additional visualizations of the model outputs at different stages of training. And finally, Section E covers additional related work.

## Appendix A    Experiment Settings

### A.1    Evaluation Metrics

(1) **MRR** evaluates a list of code snippets. The reciprocal rank for MRR is computed as $\frac{1}{rank}$, where $rank$ is the position of the correct code snippet when all code snippets are ordered by their predicted similarity to the sample query. (2) **P@K** is the proportion of the top-*K* correct snippets closest to the given query. For each query, if the correct code snippet is among the first *K* retrieved code snippets P@K=1, otherwise it is 0.

### A.2    Parsing

We build on top of the NLTK Python package for our implementation of the CCG parser. In attempt to parse as much of the datasets as possible, we preprocessed the queries by removing preceding question words (e.g. "*How to*"), punctuation marks, and some specific words and phrases, e.g. those that specify a programming language or version, such as "*in Python*" and "*Python 2.7*". For a number of entries in CSN dataset which only consisted of a noun or a noun phrase, we appended a Load verb to make it a valid sentence, assuming that it was implied, so that, for example, "*video page*" became "*Load video page*". This had the adverse effect in cases of noisy examples, where the docstring did not specify the intention or functionality of the function, and only said "*wrapper*", for example. The final dataset statistics before and after parsing are presented in Table 3

| Dataset | Parsable | | | Full | | |
|---|---|---|---|---|---|---|
| | Train | Valid | Test | Train | Valid | Test |
| CodeSearchNet | 162801 | 8841 | 8905 | 412178 | 23107 | 22176 |
| CoSQA | 14210 | - | - | 20,604 | - | - |
| WebQueryTest | - | - | 662 | - | - | 1,046 |

Table 3: Dataset statistics before and after parsing.

### A.3    Failed parses

As mentioned before, we have encountered many noisy examples and here provide samples of such examples that could not be parsed. These include cases where the docstring contains URLs, is not in English, consists of multiple sentences, or has code in it, which is often either signature of the function, or a usage example. Specific samples of queries that we couldn't parse are included in Table 5.

### A.4    Parser generalization to new datasets

In order to evaluate how robust our parser is when challenged with new datasets, we have evaluated its success rate on a number of additional datasets - containing both Python code, and code in other languages. More specifically, for a Python dataset we used CoNaLa dataset [42], using the entirety of its manually collected data, and 200K samples from the automatically mined portion. Additionally, we attempt parsing queries concerning 5 other programming languages - Go, Java,

Javascript, PHP, and Ruby. For those, we evaluated the parser on 90K for each language, taking those from CodeSearchNet dataset's training portion. The summary of data statistics, as well as evaluation results are reported in Table 4. As it can be seen, the parser successfully parses at least 62% of Python data, and 32% of data concerning other languages. From new languages, our parser is the most succesful on PHP and Javascript, achieving 43% and 41% success rate respectively.

| Language | Dataset | Original Size | Parser Success Rate |
|---|---|---|---|
| Python | CoNaLa auto-mined | 200000 | 0.62 |
| Python | CoNaLa manual train | 2379 | 0.65 |
| Python | CoNaLa manual test | 500 | 0.63 |
| Go | CodeSearchNet | 90000 | 0.32 |
| Java | CodeSearchNet | 90000 | 0.33 |
| Javascript | CodeSearchNet | 90000 | 0.41 |
| PHP | CodeSearchNet | 90000 | 0.43 |
| Ruby | CodeSearchNet | 90000 | 0.35 |

Table 4: Results of evaluation of the parser's success rate on new datasets

| URL | Example not parsed
From http://cdn37.atwikiimg.com/sitescript/pub/dksitescript/FC2.site.js |
|---|---|
| Signature | :param media_id:
:param self: bot
:param text: text of message
:param user_ids: list of user_ids for creating group or one user_id for send to one person
:param thread_id: thread_id |
| Multi-sentence | Assumed called on Travis, to prepare a package to be deployed
This method prints on stdout for Travis.
Return is obj to pass to sys.exit() directly |
| Noisy | bandwidths are inaccurate, as we don't account for parallel transfers here |

Table 5: Example queries that were not included due to query parsing errors

# Appendix B  Entity Discovery Module

To generate noisy supervision labels for the entity discovery module we used spaCy library [22] for labelling through regex matching, and Python's ast - Abstract Syntax Trees library for the static analysis labels. For the former we included the following labels: dict, list, tuple, int, file, enum, string, directory and boolean. Static analysis output labels were the following: List, List Comprehension, Generator Expression, Dict, Dict Comprehension, Set, Set Comprehension, Bool Operator, Bytes, String and Tuple. The full source code for the noisy supervision labelling procedure is available in the supplementary materials.

# Appendix C  Additional Experiments

## C.1  Unseen Entities and Actions

We wanted to see how well different models adapt to new entities and actions that were not seen during training. For that end we measured the performance of the models when broken down on queries with a different number of unseen entities (from 0 to 3+) and action (0 and 1). The results are presented in Figure 9. It can be seen that NS3 is very sensitive to unseen terms, whereas CodeBERT's performance stays the same.

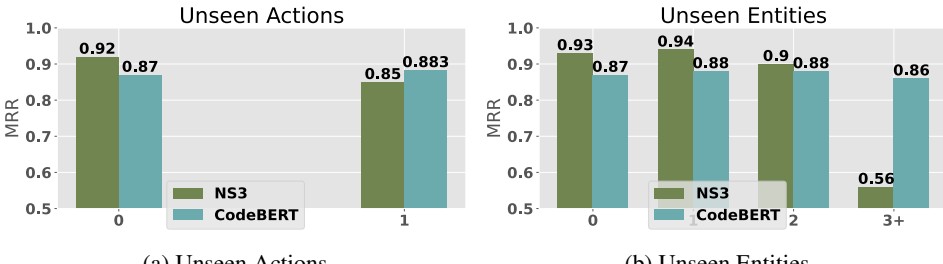

(a) Unseen Actions            (b) Unseen Entities

Figure 9: Performance of CodeBERT and NS3 models when broken down by the number of unseen entities or actions in the test queries. Evaluated on CSN test set.

## C.2   Times an Entity or an Action Was Seen

In addition to the last experiment, we wanted to measure the performance broken down by how many times an entity or an action verb was seen during the training. The results of this experiment are reported in Figure 10. For the breakdown by the number of times an action was seen, the performance almost follows a bell curve. The performance increases with verbs that were seen only a few times. On the other hand, very frequent actions are probably too generic and not specific enough (e.g. load and get). For the entities we see that the performance is only affected when none of the entities in the query has been seen. This is understandable, as in these cases an action modules don't get any information to go by, so the result is also bad. CodeBERT model in both scenarios has more or less the same performance independently of the number of times an action or an entity was seen.

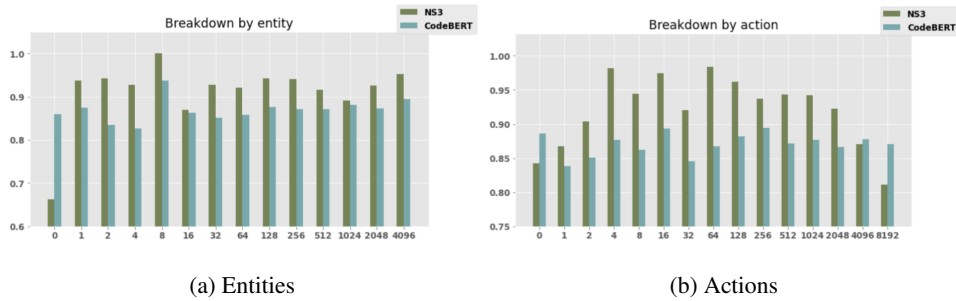

(a) Entities            (b) Actions

Figure 10: Performance of CodeBERT and NS3 models when broken down by the number of times an entity or an action was seen during the training. Evaluated on CSN test set.

## C.3   Evaluation on Parsable and Unparsable Queries

To understand whether there is a significant bias among samples that we could parse versus the ones that we could not parse, we performed additional experiment on the full test set of the CoSQA version. The results are reported in Table 6. In this evaluation, NS3 falls back to CodeBERT for examples that could not be parsed. As it can be seen, while there is some difference in performance, the overall trend of performances remains the same as before.

## Appendix D   Additional Examples

Figure 11 contains more illustrations of the output scores of the action and entity discovery modules captured at different stages of training. The queries shown here are the same, but this time they are evaluated on different functions.

**Staged execution demonstration**

In the next example we demonstrate the multiple-step reasoning. In this example we are looking at the query *"Construct point record by reading points from stream"*. When turned into a semantic parse, that query will be represented as:

| Method | CoSQA Full Test Set | | | |
|---|---|---|---|---|
| | MRR | P@1 | P@3 | P@5 |
| CodeBERT | 0.29 | 0.152 | 0.312 | 0.444 |
| GraphCodeBERT | 0.367 | 0.2 | 0.447 | 0.561 |
| NS3 | 0.412 | 0.298 | 0.452 | 0.535 |

Table 6: Mean Reciprocal Rank(MRR) and Precision@1/@3/@5 (higher is better) for different methods trained on CoSQA dataset. The performance is evaluted on the full test dataset, i.e. including both parsable and unparsable examples.

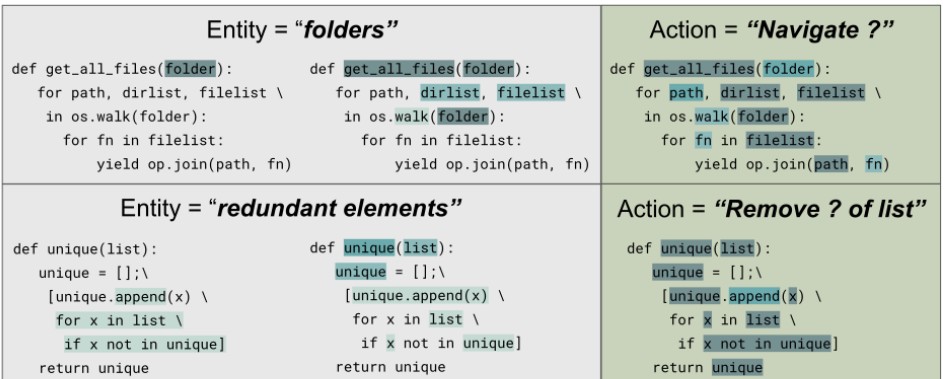

Figure 11: The leftmost column shows output scores of the entity discovery module after pretraining for the **entity** of the query. The middle column shows the scores after completing the end-to-end training. The rightmost column shows the scores of the action module. Darker highlighting demonstrates higher score.

ACTION(Construct, (None, point record),(BY, ACTION(Read, (None, points), (FROM, stream))))

After the processing, this query would be broken down into two parts:

1. ACTION(Construct, (None, point record)), and

2. ACTION(Read, (FROM, stream), (None, points))

In order for the full query to be satisfied, both parts of the query must be satisfied. Figure 12 demonstrates the outputs of the entity(Figure 12 a) and action(b) modules obtained for the query's first part, and Figure 13 demonstrates the outputs on the second part. Now if we were to replace the second sub-query with a different one, so that its parse is ACTION(Remove, (In, stream), (None, points)), that would not affect the outputs of the entity modules, but it would affect the output of the action module, as shown in Figure 14. The final prediction for this modified query would be 0.08 instead of 0.94 on the original query.

## Appendix E   Related Work

Chai et al. [8] proposes expanding CodeBERT with MAML to perform cross-language transfer for code search. In their work they study the case where the models are trained on some languages, and the then finetuned for code search on unseen languages.

Wang et al. [38] proposes combining token-wise analysis, AST processing, neural graph networks and contrastive learning from code perturbations into a single model. Their experiments demonstrate that such combination provides improvement over models with only parts of those features. This illustrates, that those individual features are complementary to each other. In a somewhat similar

```python
def from_stream(cls, stream, point_format, count):
    points_dtype = point_format.dtype
    point_data_buffer = bytearray(stream.read(count * points_dtype.itemsize))
    try:
        data = np.frombuffer(point_data_buffer, dtype=points_dtype, count=count)
    except ValueError:
        expected_bytes_len = count * points_dtype.1itemsize
        if len(point_data_buffer) % points_dtype.itemsize != 0:
            missing_bytes_len = expected_bytes_len - len(point_data_buffer)
            raise_not_enough_bytes_error(expected_bytes_len, missing_bytes_len,
                                         len(point_data_buffer), points_dtype)
        else:
            actual_count = len(point_data_buffer)
            logger.critical("Expected {} points, there are {} ({} missing)".format(
                                          count, actual_count, count - actual_count))
            data = np.frombuffer(point_data_buffer, dtype=points_dtype, count=actual_count)
    return cls(data, point_format)
```

(a) Entity outputs

```python
def from_stream(cls, stream, point_format, count):
    points_dtype = point_format.dtype
    point_data_buffer = bytearray(stream.read(count * points_dtype.itemsize))
    try:
        data = np.frombuffer(point_data_buffer, dtype=points_dtype, count=count)
    except ValueError:
        expected_bytes_len = count * points_dtype.itemsize
        if len(point_data_buffer) % points_dtype.itemsize != 0:
            missing_bytes_len = expected_bytes_len - len(point_data_buffer)
            raise_not_enough_bytes_error(expected_bytes_len, missing_bytes_len,
                                         len(point_data_buffer), points_dtype)
        else:
            actual_count = len(point_data_buffer)
            logger.critical("Expected {} points, there are {} ({} missing)".format(
                                          count, actual_count, count - actual_count))
            data = np.frombuffer(point_data_buffer, dtype=points_dtype, count=actual_count)
    return cls(data, point_format)
```

(b) Action outputs

Figure 12: Outputs of the action and entity modules on the query ACTION(Construct, (None, point record)).

manner, Guo et al. [18] proposes combining sequence-based reasoning with AST-based reasoning, and uses contrastive pretraining objective for the transformer on the serialized AST.

Additionally, both Zhu et al. [46] and Lu et al. [32] propose solutions closely inspired by human engineers' behaviors. Zhu et al. [46] propose a bottom-up compositional approach to code understanding, claiming that engineers go from understanding individual statements, to lines, to blocks and finally to functions. They propose implementing this by iteratively getting representations for program sub-graphs and combining those into larger sub-graphs, etc. On the other side, Lu et al. [32] proposes looking for the code context for the purpose of code retrieval, inspired by human behavior of copying code from related code snippets.

```
def from_stream(cls, stream, point_format, count):

    points_dtype = point_format.dtype

    point_data_buffer = bytearray(stream.read(count * points_dtype.itemsize))

    try:

        data = np.frombuffer(point_data_buffer, dtype=points_dtype, count=count)

    except ValueError:

        expected_bytes_len = count * points_dtype.itemsize

        if len(point_data_buffer) % points_dtype.itemsize != 0:

            missing_bytes_len = expected_bytes_len - len(point_data_buffer)

            raise_not_enough_bytes_error(expected_bytes_len, missing_bytes_len,
                                         len(point_data_buffer), points_dtype)

        else:

            actual_count = len(point_data_buffer)

            logger.critical("Expected {} points, there are {} ({} missing)".format(
                            count, actual_count, count - actual_count))

            data = np.frombuffer(point_data_buffer, dtype=points_dtype, count=actual_count)

    return cls(data, point_format)
```

(a) Entity outputs

```
def from_stream(cls, stream, point_format, count):

    points_dtype = point_format.dtype

    point_data_buffer = bytearray(stream.read(count * points_dtype.itemsize))

    try:

        data = np.frombuffer(point_data_buffer, dtype=points_dtype, count=count)

    except ValueError:

        expected_bytes_len = count * points_dtype.itemsize

        if len(point_data_buffer) % points_dtype.itemsize != 0:

            missing_bytes_len = expected_bytes_len - len(point_data_buffer)

            raise_not_enough_bytes_error(expected_bytes_len, missing_bytes_len,
                                         len(point_data_buffer), points_dtype)

        else:

            actual_count = len(point_data_buffer)

            logger.critical("Expected {} points, there are {} ({} missing)".format(
                            count, actual_count, count - actual_count))

            data = np.frombuffer(point_data_buffer, dtype=points_dtype, count=actual_count)

    return cls(data, point_format)
```

(b) Action outputs

Figure 13: Outputs of the action and entity modules on the query ACTION(Read, (FROM, stream), (None, points)).

```python
def from_stream(cls, stream, point_format, count):
    points_dtype = point_format.dtype
    point_data_buffer = bytearray(stream.read(count * points_dtype.itemsize))
    try:
        data = np.frombuffer(point_data_buffer, dtype=points_dtype, count=count)
    except ValueError:
        expected_bytes_len = count * points_dtype.itemsize
        if len(point_data_buffer) % points_dtype.itemsize != 0:
            missing_bytes_len = expected_bytes_len - len(point_data_buffer)
            raise_not_enough_bytes_error(expected_bytes_len, missing_bytes_len,
                                         len(point_data_buffer), points_dtype)
        else:
            actual_count = len(point_data_buffer)
            logger.critical("Expected {} points, there are {} ({} missing)".format(
                                                count, actual_count, count - actual_count))
            data = np.frombuffer(point_data_buffer, dtype=points_dtype, count=actual_count)
    return cls(data, point_format)
```

Figure 14: Outputs of the action module on the modified query ACTION(Remove, (IN, stream), (None, points)).