# OpenReview forum: "NS3: Neuro-symbolic Semantic Code Search"
_NeurIPS.cc/2022/Conference — NeurIPS 2022 Accept_

### Official Review · Reviewer_SyPM · 2022-07-09

**Rating:** 7
**Confidence:** 4
**Soundness:** 3 good
**Presentation:** 2 fair
**Contribution:** 3 good

**Summary:**

The paper introduces a novel method for semantic code search using neural module
networks. The layout of the network is produced from a semantic parse of the
query. There are two types of modules: entity discovery modules and action
modules. Entity discovery modules correspond to the nouns in the semantic parse
and each such module tries to discover the given entity (noun) of the query in
the code (they assign a relevance score to each code token).

Action modules correspond to the verbs in the query. Each action module receives
only part of the full query with the last entity argument masked, and tries to
discover the masked entity (estimate its relevance scores). The intuition is
that if the code snippet indeed corresponds to the query, then the action module
should be able to estimate the relavence scores of the masked entity based on
the code and the rest of the query. Nested action modules are flattened and
their scores are multiplied together.

The final score which measures the relatedness of the code snippet to the query
is computed by taking the normalized dot product of the relevance scores of an
entity computed by its entity discovery module and by the action module in which
it was masked. If there are multiple such scores (because there are multiple
action modules) they are multiplied together.


**Questions:**

Why is batching hard for the model? (it was mentioned in line 237)

It was mentioned that the performance of the model peaks at depth=1. Could that
be because of how compositional queries are handled (as described in lines
219-228)?

I found Figure 4 confusing. Why are both "Load 0" and "Load from" present in the
figure? On Figure 2 we just have "Load from" for the same action module and
query. Also, there is just one unmasked entity and it's without a preposition, so maybe
"Load 0" would be appropriate?


**Limitations:**

I cannot think of any limitations which are not addressed.


**Strengths And Weaknesses:**

## Strengths

The paper is an interesting and novel application of neural module networks to
semantic code search and improves the state-of-the-art results.

## Weaknesses

I think the main problem with the paper is that some parts are hard to
understand in their current form. The introduction could be more concrete with
information incorporated from the caption of Figure 2 and Section 3. For
example, I think that the introduction should mention the intuition behind
action and entity discovery modules, and how that is used to compute the
relatedness score.

I found the second paragraph of Section 3.3 especially hard to understand, I
think it should be elaborated. One reason for my confusion was the dual role of
prepositions: they belong to the entities, but we embed them with the verbs. I
still don't understand what happens when we have multiple entities as the input
dimension of the transformer would change depending on the number of entities.
Also, the dimensions, and what is concatenated to what and in which direction is
not clear.

Line 208: the code token embedding should be $c^j_k$ and not $t_k$.

Line 246: ", Section 4.1," is somewhat confusing, "later in this Section" would be better

Lines 268-277: some of the methods are not cited.

I find it odd that there is a "Background" and a "Related work" section, as
the Background already discusses the limitations of related work.

---

> ### Author Response · Authors · 2022-08-03
> **Clarity and Details of the Method**
>
>
> We are grateful for your positive comments and thoughtful suggestions. We are pleased to hear that you think the proposed method is interesting and novel. We understand the concerns you raised about the clarity of the writing. We have fixed typos, added citations, and revised the paper to expand on method ideas in the introduction, and improve the clarity of Section 3.3. We include more discussion about the intuition behind the two modules and the high-level ideas of how the module's outputs are used to compute relatedness scores. Sec 3.3 is revised to elaborate more on the module’s implementation details. Please see our general response for the detailed list of changes made to the paper. We hope that you find the overall quality of the paper has improved with this revision.
>
> In this response, we clarify implementation details about the action module (Sec 3.3) and address questions about batching and performance at different depths.
>
> ### **Clarification of the details of the action module (Sec 3.3) (Weakness 2 & Q3)**
>
> - __Modeling “preposition”:__ For each entity, there can be zero or one preposition, but each verb may have multiple entities with prepositions. Here, Load 0 indicates that there is no preposition associated with “all tables”. This design decision becomes more clear in a scenario where both entities have an associated preposition, e.g. “Load from table to file.”, In that case, the entity “table” would be matched with “load from”, and entity “file” would be masked, and the input would only contain the embedding for “load to”. We are sorry for causing confusion about this, in Figure 2 we truncated that part for brevity, and also to avoid overloading the figure with too much detail.
> - __Handling multiple entities & confusion in Figure 4:__ We handle multiple entities by having distinct copies of the module with the corresponding dimensionality for different numbers of input entities (LL202-204 in the revised version of the paper). In our current implementation, these modules do not share weights, but in a future implementation, we believe it might be beneficial to change that. To form the input vector for the Action module, we concatenate the code token embedding (dim=768), and, depending on the number of inputs one or more join embeddings of verb and preposition (dim=7) with the corresponding output score of the entity discovery module (dim=1). So, the final dimensionality of the input vector to the action module is 768 + 8*num_inputs.
> - __Updates to the Method description in Section 3.3:__ The details you queried about were missing from the paper, so we have updated Section 3.3 to include information about handling multiple inputs to the action module, and provide a clearer explanation behind our handling of prepositions.
>
> ### **Why is batching hard for the model? (Q1)**
>
> This is because our model has dynamic architecture on different query-candidate pairs since we use the semantic parse of the query to construct the layout of the network. As a result, we have a different network architecture from one query to another, Thefore, the computations for both forward and backward passes for every example are different.
>
> ### **Performance of our method at different query depths (Q2)**
>
> Thank you for raising this great question! We agree that it is interesting to investigate why our method performs worse on queries of larger depths (e.g., depth=2/3). A related observation is that the baselines perform similarly on all depths, which shows that there is no significant difference in terms of difficulty on these different subsets of queries. One hypothesis is that more compositional queries are also harder to parse for the parser, having a higher chance of ending up with a noisy or incorrect parse among longer queries. This in turn affects our model both during learning and inference. We didn’t validate this hypothesis via experiments and plan to do this in the final version.
>
> ### **The "Background" section makes the "Related work" section redundant (Weakness 3)**
>
> Our intention for the background section was to provide preliminary information and formal details, not just limitations. We are grateful for pointing out this inconsistency and will work to remove the redundancy between the two sections in the paper.

---

> > ### Comment · Reviewer_SyPM · 2022-08-07
> > **Thank you for your thoughtful answers**
> >
> > Thank you for your clarifications, especially that the modifications of the paper were easy to see in blue.
> >
> > I also think that the evaluations got much better with the additional experiments and information. What could make the paper even better would be to figure out why the method performs worse with deeper queries and maybe improve the method.
> >
> > Two small observations:
> > - maybe it would be helpful to mention explicitly in Section 3.3 that for masked tokens, only the verb-preposition embedding is included, and the entity discovery module embedding is not. It can of course be deduced, but I think it could be helpful.
> > - there is a new typo on line 32: loosing => losing
> >
> > I increased my score to Accept.

---

> ### Author Response · Authors · 2022-08-05
> **Follow up to our Response**
>
> Dear Reviewer,
>
> We wanted to follow up to check whether our responses below have addressed the concerns you have specified in your review. To briefly summarize the points covered in our response:
>
> - We have addressed your questions about the model, and implementation.
> - We have followed your suggestions for making the paper clear and have revised the Introduction and Section 3.3. for that purpose.
> - We have addressed your questions about the model performance behavior on queries of larger depth.
>
> Do not hesitate to let us know if you have any follow-up questions that we could further answer.
>
> Best regards,
>
> Authors

---

### Official Review · Reviewer_dTG5 · 2022-07-10

**Rating:** 5
**Confidence:** 4
**Soundness:** 2 fair
**Presentation:** 3 good
**Contribution:** 3 good

**Summary:**

The paper proposes the NS3 (Neuro-Symbolic Semantic Search) model, which breaks down the query into small phrases using the Categorial Grammar-based semantic parse module, which can better understand the compositional and longer queries.

To identify the similarity of each query and code snippet, the NS3 model uses two types of neural models, the entity discovery module and the action module. The entity discovery module uses a transformer encoder model and a two-layer MLP to identify the entities and their relevances. RoBERTa model initialization and noisy supervision training are applied in its self-supervised pretraining phase. The Action module architecture is similar to the entity discovery module. It estimates the action similarity through the prediction of the masked entity embedding. The module can be pre-trained with a mask-predict process for the masked entity. After pretraining, the model performs end-to-end fine-tuning for two modules.

The experiments on the CSN and CoSQA datasets show the superiority of the proposed model over the baseline methods on the parsable samples. Furthermore, the ablation study validates the effectiveness of the pretraining and investigates different score normalization methods and similarity metrics.


**Questions:**

1. Could you elaborate on the influence of the parsable sample selection on the dataset? For example, are the long sentences and samples hard-to-understand remained?

2. Is there any sample that can validate the NS3 model mimics the staged reasoning for SCS?


**Limitations:**

In the experiments, according to my understanding, only parsable data are used in the evaluation. I would suggest using all the data in the experiments to verify if the proposed method still helps in improving the SCS performance. In other words, it is not fair to compare with other baselines and models using only the dataset bias towards your method.

**Strengths And Weaknesses:**

**Strengths**:

1. The proposed NS3 model utilizes the Categorial Grammar-based(CCG) semantic parser to comprehend the semantic structure of the query better and combines the Transformer-based neural model to capture semantic information of the query text, which is an interesting idea.

2. The experiment validates the effectiveness of the proposed modules and the pretraining strategy. Furthermore, the proposed model achieves noticeable improvements over the baseline models in the parsable samples.


**Weakness**:

1. The proposed model could not operate properly in arbitrary natural language scenarios that 60% of the CSN dataset and 30% of the CoSQA dataset records are not parsable.

2. According to Line 58, the author claims the model mitigates the challenges of encoding long texts and mimics the staged reasoning for SCS. However, according to Figure 5(a), the NS3 model is not significantly improved in the deeper query situation (D=3+). The model performs better on the query with a simple semantic structure (D=1), which is inconsistent with the initial claim.

3. The unparsable issue restricts the quantity and quality of the experimental data. According to Table 1 and [1], the MRR of the GraphCodeBert model is higher on the parsable dataset (0.812 vs. 0.692). The parsable data may be easier to comprehend through the NS3 model, making the experiment comparison unfair.
Reference:

[1] Guo, D., Ren, S., Lu, S., Feng, Z., Tang, D., Liu, S., ... & Zhou, M. (2020). Graphcodebert: Pre-training code representations with data flow. arXiv preprint arXiv:2009.08366.

---

> ### Author Response · Authors · 2022-08-03
> **Part 2: Inconsistent claim, and staged reasoning examples**
>
> ### **Inconsistent claim in the paper, as the model’s performance does not improve on deeper queries  (Weakness 2)**
>
> Thank you so much for raising this crucial point. After careful consideration, we have updated the wording in the paper so that the claims reflect more precisely the strengths of our approach. We hope that you find our changes have made our submission better. Specifically, our concern is about the inability of existing models to obtain a representation of text that remains faithful to the details and query’s semantics. We demonstrate that NS3 achieves such faithfulness to details in the study in Figure 7, where we replace a single entity or a single action in samples and show that NS3 is more sensitive to this small change in semantics. We also have included additional experiments in the Appendix (Figures 9 and 10), which show that CodeBERT’s performance is roughly the same even when dealing, with unseen actions and unseen entities, which confirms again that individual unknown actions or entities have low importance in the CodeBERT model while making a prediction, thus violating the faithfulness of the representation.
>
> We suspect that the performance drop in longer queries could be due to the fact that more compositional queries are also harder to parse for the parser, having a higher chance of ending up with a noisy or incorrect parse among longer queries. This in turn affects our model both during learning and inference.
>
> Please take a look at our general comment on the claim about handling longer texts for additional information and broader discussion.
>
> ### **Is there any sample validating that the NS3 model mimics the staged reasoning for SCS? (Q2)**
>
> Thank you for raising this important point, we agree that including more evidence of NS3’s reasoning steps will help improve the paper. We have updated Appendix, Section D with a demonstration of performance on an example with multiple steps.

---

> > ### Comment · Reviewer_dTG5 · 2022-08-10
> > **Thanks for clarifications**
> >
> > Thank you for the explanations and the additional experiments to resolve my major concerns. The paper now is much better with details to support the claim. I increased my score to 5.

---

> ### Author Response · Authors · 2022-08-03
> **Parser, and potential biases introduced by evaluation on parsable examples**
>
> Thank you so much for providing a careful assessment of our paper. We are happy to hear that you found our approach of combining Transformers with semantic structure from the CCG parser interesting. We hope to have addressed some of your concerns in the general comment to all reviewers, as well as in the comments below.
>
> ### **Performance comparison on the full test set (i.e., including the unparsable instances) (Limitation 1 & Weakness 3)**
>
> Thank you for this thoughtful suggestion! We agree that reporting results on the full test set (i.e., including unparsable instances) is helpful and important. For the CoSQA dataset, we have performed experiments evaluating the model performances on both parsable and unparsable parts of the test portion of the dataset. The results are shown in the Table below. As you can see, while there is some shift in performance on the full dataset, we still see improvement in the application of NS3. This table has also been added to Appendix, section C.3.
>
> |Method|MRR|P@1| P@3|P@5|
> |---------|------|-----|-----|-----|
> |CodeBERT| 0.29 | 0.152| 0.312 | 0.444 |
> |GraphCodeBERT| 0.367 | 0.2 | 0.447 | 0.561 |
> |NS3| 0.412 | 0.298 | 0.452 | 0.535 |
>
> Furthermore, we want to clarify what we believe to be a misunderstanding of the experimental setups in our Table 1 and GraphCodeBERT paper. In our work, we follow the setup used in CodeBERT, which includes ranking the correct code snippet among 999 distractors (LL263-267). However, in GraphCodeBERT both the dataset, and the experimental setup are a little different. Firstly, they remove what they consider to be noise from the dataset, so they operate on a slightly different set of queries. Next, instead of using 999 distractors, they evaluate against a larger number of distractors, which explains why the MRR scores reported in their paper are generally lower (0.692) than those that are reported by CodeBERT and us (0.812). But the scores that we have obtained for CodeBERT on just the parsable data in CodeSearchNet are comparable to those reported in the CodeBERT paper (0.873 vs 0.868), which suggests that the parsable examples are not significantly easier for the models.
>
> ### **Applicability of the method (parser) in arbitrary natural language scenarios (Weakness 1)**
>
> While we agree that the parser poses a limitation to the applicability of our method, we want to stress that even the current implementation of the parser has demonstrated the ability to generalize outside of the dataset and the programming language for which it was created. In the paper we were able to apply the parser to CoSQA dataset without modifications, having developed it using only the training portion of the CodeSearchNet dataset. Additionally, in the table below we demonstrate the evaluation of the parser on a new dataset of Python code and English queries, as well as other datasets of 5 different programming languages and their corresponding English queries. As it can be seen from the results below, in all these cases the parsable portion of the queries is non-trivial, and our model has the potential of offering a measurable improvement on those queries. We invite you to refer to our general comment on the topic of parser for a broader discussion of the matter.
>
> Finally, we want to highlight, that it is always possible to output the results of the first-stage ranking model, CodeBERT in our case, if the query could not be parsed, still getting the boost in performance on the queries that can be parsed. This analysis has been added to Appendix, Section A.4, we also include the summary of the evaluation results table below for your convenience.
>
> |Language | Dataset | Parser Success Rate|
> |------------|----------|-------------------------|
> |Python| CoNaLa auto-mined | 0.62 |
> |Python| CoNaLa manual (train) | 0.65 |
> |Python| CoNaLa manual (test) | 0.63 |
> |Go| CodeSearchNet | 0.32 |
> |Java| CodeSearchNet | 0.33 |
> |Javascript| CodeSearchNet | 0.41 |
> |PHP| CodeSearchNet | 0.43 |
> |Ruby| CodeSearchNet | 0.35 |
>
> ### **Elaboration on the influence of the parsable sample selection on the dataset? (Q1)**
>
> First of all, we would like to highlight the point from above, that the trends of model performances remain the same when evaluating on all examples. In addition, Appendix Section A.3 and Table 4 have a discussion on noisy samples in the datasets. To summarize briefly, during manual observations of failed parses we have found that a lot of the sentences that could not be parsed were not in fact queries of code, but rather things like URLs, signatures of functions, or comments unrelated to the semantics of the functions. And finally, we want to stress again that for the large portions of the dataset that we could parse we improve the performance by a significant margin, so combining our method as a second step on top of another base method, e.g. CodeBERT, whenever parses are available is a simple way to garner that improvement.

---

> ### Author Response · Authors · 2022-08-05
> **Follow up to our Response**
>
> Dear Reviewer,
>
> We wanted to follow up to check whether our responses below have addressed the concerns you have specified in your review. To briefly summarize the points covered in our response:
>
> - We have addressed your concern about the mismatch between our claim and experimental results, clarified the corresponding claim, and updated the paper.
> - We have pointed out a small misunderstanding of the experimental setup in one of our baselines which led to an incorrect conclusion about the prevalence of the sample selection biases in our evaluation. Additionally, by your suggestion, we have provided experimentation results on the full dataset to demonstrate the performance gain is maintained in the scenario without any biases.
> - We have answered your questions about the parser and provided additional experimentation that shows how our parser can successfully generalize to new scenarios.
>
> Do not hesitate to let us know if you have any follow-up questions that we could further answer.
>
> Best regards,
>
> Authors

---

> ### Author Response · Authors · 2022-08-08
> **Additional Questions?**
>
> Dear Reviewer,
>
> With the nearing end of the discussion period, we wanted to reach out again and check if you were satisfied with our answers below, as well as see if there are any additional questions or concerns we could address for you.
>
> Best Regards,
>
> Authors

---

> ### Author Response · Authors · 2022-08-09
> **Last Check-In**
>
> Dear Reviewer,
>
> We wanted to use this last chance to check in and see whether you found our following clarifications and updates satisfactory:
>
> - In response to your concern about the mismatch between one of our claims and the experimental results, we have clarified the corresponding claim according to the results and updated the paper. Please see our previous comment to you, as well as the general comment on this matter for details.
> - In response to your conclusion about the sample selection bias in our evaluations, we have pointed out a misunderstanding of the experimental setup in one of our baselines which led to the incorrect conclusion.
> - Additionally, following your suggestion, we have provided experimentation results on the full dataset to demonstrate the performance gain is maintained in the absence of sample selection.
> - We have answered your questions about the parser and provided additional experimentation that shows how our parser can successfully generalize to new datasets and new programming languages.
>
> We encourage you to refer to our previous comments for details on these points.
>
> Best regards,
>
> Authors

---

### Official Review · Reviewer_FqUC · 2022-07-11

**Rating:** 5
**Confidence:** 4
**Soundness:** 3 good
**Presentation:** 3 good
**Contribution:** 3 good

**Summary:**

The paper studies the problem of retrieving code snippets given textual queries (called semantic code search). The work is motivated by language models’ limitations on encoding longer and compositional text (which I question a bit about, see my comments below). The authors propose a neural module network (called NS3) and introduce a modular workflow according to the semantic structure of the query. More specifically, NS3 contains two types of neural modules, entity discovery module, and action module, to estimate the semantic relatedness of code tokens and entity mentions and actions in a query separately. It decomposes the task into multiple steps of matching each data entity and action in the query. The authors demonstrate the effectiveness of the proposed method on several code search benchmarks and show that their method outperforms some strong baselines in some settings (on which I’m a bit confused).


**Questions:**

Line 246-248: I’m confused about the evaluation data statistics. Is it a common practice to exclude unparsable examples for evaluation? It looks like that the authors did that only because of the limitations of their method (only taking parsable queries). It would be helpful to provide data statistics both before (original) and after parsing in Appendix A.2 (Table 3).

Line 254: Did you randomly select 5k and 10k examples? If they are randomly chosen did you report results on multiple randomly selected examples?

Line 264-266: Why did you focus on the two-step evaluation? Again, do CodeBERT/GraphCodeBERT themselves (not by yourselves) also report results in this setting? It seems that they can be applied in a single-step setting. Why not evaluate your method in that setting too without depending on CodeBERT’s first step predictions (then only applying your method to rerank the top 10 CodeBERT predictions)? As you mentioned in Line 303-305, the highest possible results you could get with this evaluation strategy is kind of low (74% on CoSQA…)...

Line 307- Fig. 5: Do you have any explanations about why GraphCodeBERT performs much worse than CodeBERT in many cases? Is it the strongest baseline you compare with?


**Strengths And Weaknesses:**

Strengths:

The problem of semantic code search is very interesting. It is easy to follow the writing of the paper. The authors compare the proposed methods with other works and show strong performance on multiple code search benchmarks.

Weaknesses:

The authors mention that one of the main motivations of the work is because language models struggle with encoding long and compositional text. I’m a bit suspicious. Text queries as examples shown in the paper (but not just these examples, generally speaking) are not very long and complicated. Some of them are even too simple and ignore some details, which leads to mismatching between the query and code (which could be the real challenge). Language models are able to encode much more complex and longer texts than these examples in the paper….

According to what the authors say in lines 265-266, CodeBERT and many other simpler approaches can be fed examples in batches, which make them much faster in retrieval settings. In real-world retrieval settings, efficiency is also a very important consideration. In this case, the proposed method NS3 is less attractive (also depending on other methods such as CodeBERT). This makes NS3 look more like a re-ranking model instead of a fully actionable code retrieval model.

The query parser implementation seems to be a lot of human work (e.g., building a vocab of action and entity words).

---

> ### Author Response · Authors · 2022-08-03
> **Part 2: parser and more questions about evaluation**
>
> ### **The query parser implementation seems to be a lot of human work (Weakness 3)**
>
> The parser rules were developed by us without any prior experience with CCG parsing. It was performed through a few iterations of manual assessment of the parsing results obtained on the CodeSearchNet dataset’s training set, and adjusting the parsing rules accordingly. The majority of the process took roughly two weeks. We want to note that we did not perform further changes to the parser depending on the performance of the NS3 model on the development or testing sets of either dataset. The resulting parser was robust enough to be applicable to CoSQA dataset without modifications. In addition, we have performed more experiments evaluating the parser on both queries concerning Python code, as well as code in five other programming languages (Go, Java, Javascript, PHP, Ruby). As it can be seen, the parser can parse 62% of the queries about Python, and at least 32% of queries about other languages, with PHP being the highest at 42%. Please refer to our comment to all reviewers for a broader discussion on the generality of the parser.
>
> |Language | Dataset | Parser Success Rate|
> |------------|----------|-------------------------|
> |Python| CoNaLa auto-mined | 0.62 |
> |Python| CoNaLa manual (train) | 0.65 |
> |Python| CoNaLa manual (test) | 0.63 |
> |Go| CodeSearchNet | 0.32 |
> |Java| CodeSearchNet | 0.33 |
> |Javascript| CodeSearchNet | 0.41 |
> |PHP| CodeSearchNet | 0.43 |
> |Ruby| CodeSearchNet | 0.35 |
>
> ### **Did you randomly select 5k and 10k examples? (Q2)**
>
> 5K and 10K sample scenarios were sampled randomly, and for those experiments, we do not report the average over multiple subsamples.
>
> ### **Why did you focus on the two-step evaluation? (Q3)**
>
> Regarding us performing two-stage retrieval for evaluation, we want to mention that two-stage retrieval is not uncommon in information retrieval with a less precise, but faster method filtering a large number of examples, and a second, slower but more precise, method re-evaluating most likely candidates to make the final decision. You are correct in your understanding that both CodeBERT and GraphCodeBERT can be used in just 1 step. But as we see from our results in Tables 1 and 2, there is an improvement that can be obtained by applying NS3 as a second stage.  The reason we did not run our model in just 1 stage is resource efficiency. Our model uses dynamic architecture (LL142-163), which makes it harder to batch examples together and process multiple examples at once because from one example to another we end up with a different model architecture.
>
> ### **Why does GraphCodeBERT perform much worse than CodeBERT in Figure 5? (Q4)**
>
> From our results there is no way to conclusively decide whether GraphCodeBERT is a stronger baseline than CodeBERT; in particular, in Figure 5 that you mention, GraphCodeBERT is stronger on CoSQA.

---

> ### Author Response · Authors · 2022-08-03
> **Inaccurate claim; clarifications about runtime and evaluation**
>
> We want to thank you for your insightful comments and suggestions. We have attempted to address a number of concerns that you have expressed in the general comments to all reviewers, and hope that you find our answers satisfying. Additionally, we have updated the paper to address the issue with regards to the correctness of our claim of the model performance on long and compositional text, we have rephrased the claim to focus more on the faithful representation of the details in the query.
>
> ### **Inaccurate claim on the advantage of the proposed method (Weakness 1)**
>
> As you have correctly pointed out, one of the main claims in the paper, regarding encoding longer queries, was not backed by the experimental results. We have updated the paper to make the claim more true and correct, by specifying that we are interested in a representation of the query that is faithful to the details and its semantics. In other words, details or parts of the query are not misrepresented or omitted. We believe that from our experiments in Figure 7, it can be observed that NS3 is in fact obtaining a more faithful representation since it is more sensitive to small, but semantically important changes to the query. We discuss this more broadly in our general comment on the topic.
> We are also working to include longer and more compositional examples in the paper and will update the draft accordingly in the next couple of days.
>
> ### **Runtime and best usage scenarios (Weakness 2)**
>
> There is some tradeoff of efficiency for performance during inference, but we believe it is not prohibitively large for real-world retrieval scenarios. We have performed a small benchmarking experiment, and on an NVIDIA Tesla V100 GPU CodeBERT takes 0.25 seconds to process 1000 samples, and NS3 then takes 0.45 seconds to perform the re-ranking of the top 10 samples. The overhead of training a new neural model is a one-time effort, and we spent between 10 and 15 hours training the model on an NVIDIA Tesla V100 GPU.
> Your comment about NS3 being used as a re-ranker is on-point, it is indeed the use case we have in mind for our model (LL237-242; LL263-267). For that reason we have also included a similarly evaluated re-ranking baseline with a different model in the second stage, these results are referred to as “GraphCodeBERT*” in Tables 1 and 2.
> Finally, we wanted to bring your attention to the fact that NS3 can be used on top of any base retrieval method for filtering the pool of candidates, not necessarily just CodeBERT.
>
> ### **Evaluation setup and results on the full test set (Q1)**
>
> We have excluded portions of the data that we could not parse because it is always possible to fall back to an end-to-end model, such as CodeBERT, in cases when the parser fails. Our parser can successfully parse a measurable portion of both datasets, about 40% of CSN and 70% of CoSQA, and NS3 shows a non-trivial improvement on the parsed portion, about 0.05 MRR on CSN and 0.2 MRR on CoSQA, so overall its application will be beneficial. We have also updated Appendix A.2, Table 3, to include the full dataset statistics before parsing.
>
> |Dataset|Train|Valid|Test|Full Train|Full Valid|Full Test|
> |-|-|-|-|-|-|-------------|
> |CodeSearchNet|162801| 8841| 8905| 412178| 23107| 22176|
> |CoSQA|14210 |-| -| 20,604 |-| -|
> |WebQueryTest|- |- |662| - |- |1046|
>
> In addition, we have performed experiments on the full CoSQA test set, i.e. including both parsable and unparsable examples. As it can be seen from the results table below, while there is a shift in performance, the overall trends of the performance remain the same, and NS3 still offers improvement when used on top of CodeBERT. We have also updated Appendix C.3 accordingly, to include this experiment.
>
> |Method|MRR|P@1| P@3|P@5|
> |---------|------|-----|-----|-----|
> |CodeBERT| 0.29 | 0.152| 0.312 | 0.444 |
> |GraphCodeBERT| 0.367 | 0.2 | 0.447 | 0.561 |
> |NS3| 0.412 | 0.298 | 0.452 | 0.535 |

---

> ### Author Response · Authors · 2022-08-05
> **Follow up to our Response**
>
> Dear Reviewer,
>
> We wanted to follow up to check whether our responses below have addressed the concerns you have specified in your review. To briefly summarize the points covered in our response:
>
> - We have addressed your concern about the mismatch between our claim and experimental results, clarified the corresponding claim, and updated the paper.
> - We have answered your questions about the evaluation setup, and the rationale behind our setup.
> - We have provided experimentation on the full dataset to demonstrate the performance of the models without additional sample selection biases.
> - We have answered your questions about the parser and provided additional experimentation that shows how our parser can successfully generalize to new scenarios.
> - Following your suggestions, we have made some other changes to the paper, such as providing additional statistics you had requested.
>
> Do not hesitate to let us know if you have any follow-up questions that we could further answer.
>
> Best regards,
>
> Authors

---

> ### Author Response · Authors · 2022-08-08
> **Additional Questions?**
>
> Dear Reviewer,
>
> With the nearing end of the discussion period, we wanted to reach out again and check if you were satisfied with our answers below, as well as see if there are any additional questions or concerns we could address for you.
>
> Best Regards,
>
> Authors

---

> > ### Comment · Reviewer_FqUC · 2022-08-09
> > **Thanks for your clarifications**
> >
> > Thank you for your responses and clarifications! They clarify some of my questions. Based on a clearer understanding of the paper, I decided to keep my score at 5.

---

### Official Review · Reviewer_kn2F · 2022-07-11

**Rating:** 5
**Confidence:** 3
**Soundness:** 2 fair
**Presentation:** 4 excellent
**Contribution:** 2 fair

**Summary:**

This paper proposes NS3, Neuro-Symbolic Semantic Code Search. NS3 supplements the query sentence with a layout of its semantic structure, which is then used to break down the final reasoning decision into a series of lower-level decisions. NS3 outperforms baselines on CodeSearchNet and CoSQA.

**Questions:**

What are the process of engineering parsing rules and NL-action mappings? Are they engineered according to the performance on the dev set or the test set?

**Strengths And Weaknesses:**

Strengths:
1. NS3 is evaluated in two well-established benchmarks, CodeSearchNet and CoSQA, and compared against strong baselines.
2. The empirical result is very strong. The proposed method, NS3, outperforms state-of-the-art by a large margin.
3. The paper includes detailed experiments on reduced dataset settings and ablation studies.

Weaknesses:
1. NS3 requires rule-based parsing of natural languages, while other baselines, such as CodeBERT, only involve natural language models. Rule-based parsing of natural languages is complicated, language-dependent, and less scalable than language models. In `parser_dict.py` of the released parser, there are hundreds of task-specific parsing/synonym rules for natural languages. These rules are not transferrable to different natural languages (e.g. English to Chinese), and they even cannot generalize to different programming languages (e.g. Python to C++).

---

> ### Author Response · Authors · 2022-08-03
> **Parser**
>
> Thank you so much for bringing up many important points, and appreciating the care we put into designing our experimental setting and ablation studies. We agree with your concerns regarding the parser and hope to have addressed most of them below, as well as in the general comment to all reviewers.
>
> ### **Potential of the parser to generalize to new datasets and other programming languages (Weakness 1)**
> After developing the parser on the CodeSearchNet dataset’s training portion, we did not perform additional changes to parse the CoSQA dataset. In other words, the parser that was created for CodeSearchNet was robust enough to be applied to another dataset without significant modifications. In addition, we have performed a study on new datasets (Appendix A.4), to evaluate the performance of the parser on a) an unseen dataset of English queries on Python code, and b) unseen datasets of English queries for code in other programming languages (Go, Java, JavaScript, PHP, Ruby). According to this experiment, we were able to parse at least 62% of Python code queries, and at least 32% of queries for other languages, with PHP being the highest at 43%.
>
> |Language | Dataset | Parser Success Rate|
> |------------|----------|-------------------------|
> |Python| CoNaLa auto-mined | 0.62 |
> |Python| CoNaLa manual (train) | 0.65 |
> |Python| CoNaLa manual (test) | 0.63 |
> |Go| CodeSearchNet | 0.32 |
> |Java| CodeSearchNet | 0.33 |
> |Javascript| CodeSearchNet | 0.41 |
> |PHP| CodeSearchNet | 0.43 |
> |Ruby| CodeSearchNet | 0.35 |
>
> In future work, it is possible to alleviate the need for the parser altogether, by asking the users of code search to formulate their semantic queries in some formal way instead, for example, following the syntax of SQL or CodeQL queries.
>
> ### **Potential of the parser to generalize to new natural languages, e.g. English->Chinese (Weakness 1)**
>
> As you have correctly noted, generalizing to a new language, such as Chinese instead of English is infeasible for the current implementation of our semantic parser. However, most models, including CodeBERT or  GraphCodeBERT would have to be retrained from scratch for that scenario, requiring a lot of additional work, so we do not believe such a scenario imposes a limitation unique to just our approach.
>
> ### **Engineering and selection of parsing rules (Q1)**
>
> Rules for the parser were indeed created manually. In the process of rule-creation, we used the training portion of the CodeSearchNet dataset. This process was guided by a qualitative assessment of the output parses on the training portion of the CodeSearchNet dataset. We want to highlight, however, that we did not perform additional changes to the parser based on the performance of the full NS3 model on either development or test sets.

---

> > ### Comment · Reviewer_kn2F · 2022-08-10
> > **Thank you for your response**
> >
> > Thank you for addressing my concerns regarding parsing rules and providing additional experiments about the parser success rate on other datasets. However, given that the parser success rate on other CodeSearchNet datasets is still low (<=41%), a user still needs to add NL parsing rules when they use the proposed method on other datasets. Moreover, training a language model on another natural language is automatic and requires much less effort by NLP experts than writing parsing rules. So I am not changing my review scores at this time.

---

> ### Author Response · Authors · 2022-08-05
> **Follow up to our Response**
>
> Dear Reviewer,
>
> We wanted to follow up to check whether our responses below have addressed the concerns you have specified in your review. To briefly summarize the points covered in our response:
>
> - We have answered your questions about the process that went into the implementation of the parser.
> - We have provided additional experimentation that shows how our parser can successfully generalize to new scenarios.
>
> Do not hesitate to let us know if you have any follow-up questions that we could further answer.
>
> Best regards,
>
> Authors

---

> ### Author Response · Authors · 2022-08-08
> **Additional Questions?**
>
> Dear Reviewer,
>
> With the nearing end of the discussion period, we wanted to reach out again and check if you were satisfied with our answers below, as well as see if there are any additional questions or concerns we could address for you.
>
> Best Regards,
>
> Authors

---

> ### Author Response · Authors · 2022-08-09
> **Last Check-In**
>
> Dear Reviewer,
>
> We wanted to use this last chance to check in and see whether you found our following clarifications and updates satisfactory:
>
> - We have answered your questions about the parser and provided additional experimentation that shows how our parser can successfully generalize to new datasets and new programming languages.
>
> We encourage you to refer to our previous comments for details on these.
>
> Best regards,
>
> Authors

---

### Author Response · Authors · 2022-08-03
**The Procedure of Creation of the Parser and Discussion on its Performance**

Rules for the parser were created manually. In the process of rule-creation, we used the training portion of the CodeSearchNet dataset. This process was guided by a qualitative assessment of the output parses on the training portion of the CodeSearchNet dataset.
We want to highlight, however, that we did not perform additional changes to the parser based on the performance of the full NS3 model on either development or test sets, and neither did we make changes to the parser to parse the CoSQA dataset in addition to CodeSearchNet later. In other words, the parser that was created for CodeSearchNet was robust enough to be applied to another dataset without significant modifications.

In addition, we have performed a study on new datasets (Appendix A.4), to evaluate the performance of the parser on a) an unseen dataset of English text and Python code, and b) unseen datasets of English text and code in other programming languages (Go, Java, JavaScript, PHP, Ruby). According to this experiment, we were able to parse at least 62% of English queries on Python data, and at least 32% of queries for other languages, with PHP being the highest at 43%. Full results are presented in the Appendix, below is the summary:

|Language | Dataset | Parser Success Rate|
|------------|----------|-------------------------|
|Python| CoNaLa auto-mined | 0.62 |
|Python| CoNaLa manual (train) | 0.65 |
|Python| CoNaLa manual (test) | 0.63 |
|Go| CodeSearchNet | 0.32 |
|Java| CodeSearchNet | 0.33 |
|Javascript| CodeSearchNet | 0.41 |
|PHP| CodeSearchNet | 0.43 |
|Ruby| CodeSearchNet | 0.35 |

We also want to note, that while undeniably the parser can be improved, our manual assessment of failed parses shows that a lot of those sentences do not in fact represent proper queries, and are noisy instances in the datasets. Among such instances we have discovered URLs, function signatures, unrelated comments to the semantics of the function, and others. We discuss this and provide specific examples in Appendix A.3, Table 4.

And finally, we performed an evaluation on the full testing portion of the CoSQA dataset, to demonstrate that NS3 still improves the performance when compared to CodeBERT and GraphCodeBERT baselines. This experiment is discussed in Appendix, Section C.3 and Table 6. Below we present a summary of the table.
|Method|MRR|P@1| P@3|P@5|
|---------|------|-----|-----|-----|
|CodeBERT| 0.29 | 0.152| 0.312 | 0.444 |
|GraphCodeBERT| 0.367 | 0.2 | 0.447 | 0.561 |
|NS3| 0.412 | 0.298 | 0.452 | 0.535 |

---

### Author Response · Authors · 2022-08-03
**Regarding Inaccurate Claim on the Advantage of the Proposed Method**

We are very grateful to the reviewers for pointing out that one of our claims, which was formulated as the ability to encode longer texts in contrast to existing models (L58), was inaccurate and inconsistent with the results. More precisely, and we have updated the wording in the paper to reflect this, our concern is about the inability of existing models to obtain a representation of the text that remains faithful to the details and query’s semantics. We demonstrate that NS3 achieves such faithfulness in the study in Figure 7, where we replace a single entity or a single action in samples and show that NS3 is more sensitive to this small change in semantics. We also have included additional experiments in the Appendix (Figures 9 and 10), which show that CodeBERT’s performance is roughly the same even when dealing with unseen actions and unseen entities, which confirms again that an individual unknown action or entity have low importance in the CodeBERT model while making a prediction, thus violating the faithfulness of the representation.
We have updated the corresponding claims in the Introduction, when discussing the advantages and evaluation results, as well as in Limitation in Section 2.2, and hope it makes the paper more clear and sound.

---

### Author Response · Authors · 2022-08-03
**Summary of our Rebuttal and Revision**

Dear Reviewers,

We want to thank you for your careful comments and valuable suggestions. We were pleased to hear that the **following strengths** of our work have caught your eye:
- **kn2F**: very strong empirical results, including detailed evaluations and ablation studies.
- **FqUC**: strong performance on an interesting problem
- **dTG5**: strong performance achieved with an interesting solution using a combination of symbolic information from CCG parse and neural approach based on Transformer
- **SyPM**: strong performance achieved on an interesting and novel application of neural module networks to a novel problem.

We also are very grateful for constructive criticisms - we have received a number of great questions and suggestions for improvement of the paper. We hope that **our answers will clear your concerns** that we have outlined below, and you will find that the overall quality of our submission has improved after making the following changes:
- **kn2F, FqUC, dTG5**: performance of the parser as well as its generalization capacity. We have updated Appendix Section A.4 to include additional evaluations of the parser on a number of new code-search datasets. The datasets are comprised of English text and code in Python, as well as 5 other programming languages (Go, Java, Javascript, PHP, Ruby).
- **FqUC, dTG5**: how parsable vs un-parsable examples skew the performance of the models. We have updated Appendix Section C.3 to include the performance evaluation for our model, CodeBERT, and GraphCodeBERT, when evaluated on the entire dataset, as opposed to just its parsable portion. We have also updated Appendix Section A.2, Table 3, to include original data statistics before parsing
- **dTG5, FqUC**: the correctness of the claim about the advantage of the proposed method for longer texts. We have rephrased the corresponding claim to specify our model focuses on the faithful representation of details of the query, and walk through experiments that evidence this.
- **SyPM**: Clarity issues in the introduction and Section 3.3. We have updated those sections to improve the clarity of the writing and added more details about implementation to avoid confusion by the reader. We have also fixed some typos.


For your convenience, **the changes to the revised paper** and supplementary material (appendix) **are emphasized with blue text**. To make it more convenient to check our appendix, we only uploaded the appendix pdf file as the supplementary material. Our previous zip file with code is still accessible in the revision history.

We thank you again for taking the time to review our paper and engage in the discussion.

Authors

---

### Meta-Review · Area_Chair_yENd · 2022-08-20

**Recommendation:** Accept
**Confidence:** Less certain

**Metareview:**

The paper studies the problem of retrieving code snippets given textual queries (NS3, Neuro-Symbolic Semantic Code Search).  The work is motivated by language models’ limitations on encoding longer and not providing a faithful explanation of their reasoning on compositional queries.  NS3 supplements the query sentence with a layout of its semantic structure, which is then used to break down the final reasoning decision into a series of lower-level decisions. NS3 outperforms baselines on CodeSearchNet and CoSQA.


Overall, the reviewers liked the motivation of the work. A lot of concerns have been raised with extensive responses from the authors. Some of these concerns remain and have to be acknowledged and clarified in the modified document. **Despite the limitations of the work, I think "accepting" this work outweighs "rejecting" it, assuming that the authors will put due diligence into improving their drafts based on their latest experiments (outlined in the author's response), along with a clear discussion of their limitations.**



Let me start with the strengths:

 - All reviewers have found the work interesting.

 - Strong empirical results: NS3 is evaluated on two well-established benchmarks, CodeSearchNet and CoSQA. The empirical result is very strong. The proposed method, NS3, outperforms state-of-the-art by a large margin.

 - The paper includes detailed experiments on reduced dataset settings and ablation studies.


Here are several points of concern that came up in the reviews:

 - NS3 requires rule-based parsing of natural languages (unlike other LM-based baselines such as CodeBERT). The difficulty of this construction ("roughly two weeks") brings up several concerns:
 	- These rules might not generalize to different programming languages (e.g. Python to C++): on this point, the authors have reported some evidence of generalization though the reviewer "kn2F" has viewed it as "low (<=41%)" and not convinced. I suggest the authors report these analyses in their main draft. I suspect these numbers need to be compared with the corresponding numbers of their end-to-end baselines.
 	- These rules might not transferrable to different **natural** languages. The authors have acknowledged this limitation. I suggest the authors be explicit about such limitations in their work.
 	- These rules might not generalize to longer queries; this is acknowledged in the author's response. I suggest the authors be upfront about these issues in their draft.




**Award:**

No

---

### Decision · Program_Chairs · 2022-09-14

Accept